# Automatic Range Adjustment of the Fluorescence Immunochromatographic Assay Based on Image Processing

**DOI:** 10.3390/s20010209

**Published:** 2019-12-30

**Authors:** Ruixin Jiang, Huihuang Wu, Jianpeng Yang, Haiyan Jiang, Min Du, Mangi Vai, Siohang Pun, Yueming Gao

**Affiliations:** 1College of Physics and Information Engineering, Fuzhou University, Fuzhou 350108, China; 2Key Lab of Medical Instrumentation & Pharmaceutical Technology of Fujian Province, Fuzhou 350108, China; 3Key Lab of Eco-Industrial Green Technology of Fujian Province, Nanping 354300, China; 4State Key Laboratory of Analog and Mixed-Signal VLSI, University of Macau, Macau 999078, Macau SAR, China; 5Department of Electrical and Computer Engineering, Faculty of Science and Technology, University of Macau, Macao 999078, China

**Keywords:** fluorescence immunochromatographic assay, point-of-care testing, automatic range adjustment, image processing, exposure time

## Abstract

As an emerging technology, fluorescence immunochromatographic assay (FICA) has the advantages of high sensitivity, strong stability and specificity, which is widely used in the fields of medical testing, food safety and environmental monitoring. The FICA reader based on image processing meets the needs of point-of-care testing because of its simple operation, portability and fast detection speed. However, the image gray level of common image sensors limits the detection range of the FICA reader, and high-precision image sensors are expensive, which is not conducive to the popularization of the instrument. In this paper, FICA strips’ image was collected using a common complementary metal oxide semiconductor (CMOS) image sensor and a range adjustment mechanism was established to automatically adjust the exposure time of the CMOS image sensor to achieve the effect of range expansion. The detection sensitivity showed a onefold increase, and the upper detection limit showed a twofold increase after the proposed method was implemented. In addition, in the experiments of linearity and accuracy, the fitting degree (*R*^2^) of the fitted curves both reached 0.999. Therefore, the automatic range adjustment method can obviously improve the detection range of the FICA reader based on image processing.

## 1. Introduction

Immunochromatographic assay (ICA) [1] combines the chromatographic assay and the specific reaction of antigen and antibody. With biomarkers as the tracer, useful information can be obtained directly or indirectly through optical instruments to achieve quantitative detection of the analyte. Due to its high sensitivity and fast detection speed, ICA has been extensively used in environmental [2], medical [3,4], and food safety detection [5,6,7,8].

There are mainly three kinds of biomarkers in the ICA applications, including nano-gold [9], quantum-dot [10,11], and fluorescent particles [12,13]. Compared with the traditional ICA using nano-gold, fluorescence immunochromatographic assay (FICA) can be used for quantitative determination of the analyte. The existing FICA research can be classified into photoelectric and image detection. Photoelectric FICA is used to detect melamine in milk [14]; the range of the reader was previously adjusted by controlling the light intensity [15]. For the FICA based on image processing, auto fluorescence multispectral imaging was used to assess the advanced glycated end product accumulation in skin [16]. This paper designed a portable FICA system based on auto fluorescence imaging. Fluorescence imaging technology has been applied in clinical trials [17].

To meet more application needs, such as clinical testing and public health, FICA requires higher sensitivity and wider detection range. There are some deficiencies in existing FICA readers. The photoelectric FICA reader uses a stepper motor to drive the strip, which takes a long time to detect, and the scanning device has a high demand for the positioning of the optical system. The FICA reader based on image processing uses the image sensor to collect the fluorescence strip image, which can improve the stability of the detection result, and has the advantages of fast detection speed and simple hardware. However, the gray level of common 8-bit image sensors limits the detection range of the instrument. The concentration of some strips is not within the detection range of the reader, which makes it impossible to extract the real data of the strips [18].

This paper pointed out the problem of insufficient detection range of the FICA reader using common image sensors and proposed a solution. The influence of the camera exposure time on the FICA strips was determined, and the detection range of the FICA reader based on image processing was expanded by controlling the exposure time of the camera.

## 2. Materials and Methods

### 2.1. Basic Principles

#### 2.1.1. Detection Mechanism of the FICA Based on Image Processing

The strip is the carrier of immunochromatographic reaction, and the fluorescent substances of the strip emit fluorescence under the illumination of the excitation light source. The relationship between the fluorescence intensity and the current value of the excitation light source [15] is as follows:(1)$$I_{f}=2.3Y_{F}\varepsilon Cl\alpha i_{X},$$
where *I*_f_ is the fluorescence intensity; *Y*_F_ is the fluorescence efficiency; *ε* is the molar absorption coefficient; *l* is the optical path of the excitation light distance from the sample; *C* is the concentration of the tested sample; *i*_x_ is the current value of the excitation light source; *α* is the proportional coefficient of the excitation light intensity and the current value; and *Y*_F_, *l*, *ε*, *C*, and *α* remain unchanged under the condition that the same strip is detected by the reader. According to eq (1), the fluorescence intensity is linearly related to the current value of the excitation light source under the same concentration of the analyte. Therefore, the fluorescence intensity can be changed by controlling the current value *i_x_* of the excitation light source.

In sensitometry, photometric exposure is defined as the product of the light intensity of visible light produced on the surface of the photosensitive material and the corresponding exposure time [19]. Photometric exposure can be expressed as
(2)H=Et,
where *H* is the photometric exposure (lux × s); *E* is the light intensity (lux); and *t* is the exposure time (s). In the process of detecting the strip, the ratio of the characteristic value of test line (T line) to that of control line (C line) is taken as the final detection result. Therefore, the detection result for the camera is essentially the ratio of the photometric exposure of T line to that of C line.

In a confined space, *E*≈*I*_f_. Combining with equations (1) and (2), the photometric exposure of T line can be obtained as follows:(3)$$H_{T}=2.3Y_{F}\varepsilon C_{T}l\alpha i_{T}t.$$

Similarly, the photometric exposure of C line is
(4)$$H_{C}=2.3Y_{F}\varepsilon C_{C}l\alpha i_{C}t,$$
therefore, the detection result *T*/*C* is
(5)$$\frac{H_{T}}{H_{C}}=\frac{2.3Y_{F}\varepsilon C_{T}l\alpha i_{T}t}{2.3Y_{F}\varepsilon C_{C}l\alpha i_{C}t}=\frac{C_{T}i_{T}}{C_{C}i_{C}},$$
where *H*_T_ and *H*_C_ are the photometric exposure on the camera of the fluorescent substances in T line and C line, respectively; *C*_T_ and *C*_C_ are the concentrations of the fluorescent substances in T line and C line, respectively; and *i*_T_ and *i*_C_ are the current values of the excitation light sources of T line and C line, respectively. The strip was illuminated during detection by the same excitation light source; thus, *i*_T_ = *i*_C_. Therefore, the final detection result *T*/*C* is equal to *C*_T_/*C*_C_, which is only related to the concentrations of the fluorescent substances in T line and C line and has nothing to do with the exposure time of the camera.

Therefore, controlling the exposure time to adjust the detection range of the reader has no effect on the final result *T*/*C* of the FICA strip. Thus, this method can be applied to FICA based on image processing.

#### 2.1.2. Mechanism of Exposure Time Adjustment

LED of type TH-UV365T3WA-3535 was selected as the excitation light source of the reader and was focused by the convex lens to generate uniform planar light in a certain circular range. The current value of the excitation light source was set as the rated current to detect the variation rule of the image gray value with the exposure time. As shown in Figure 1, curve fitting was performed between the gray value and the exposure time. The fitting degree (*R*^2^) of the fitted curve reached 0.999.

The linear relationship is
(6)G=0.547t+3.480,
where *G* is the gray value of the image and *t* is the exposure time. The gray value of the image is proportional to the exposure time when the exposure time is less than 400 ms. Therefore, the exposure time should be adjusted within the linear range in the detection process to ensure the accuracy of the detection results.

### 2.2. Methods

#### 2.2.1. Devices

The image acquisition device of the reader consists of LED excitation light sources with a wavelength of 365 nm, a complementary metal oxide semiconductor (CMOS) camera, and a filter with transmission wavelengths of 365 and 610 nm. Figure 2 shows the device. The excitation light was illuminated on a C line and a T line, producing the emitted light with a wavelength of 610 nm. The light enters the camera through the filter and is transmitted to the upper computer, which processes the image. Figure 3 illustrates the process.

In order to successfully segment the image, the original image must be denoised due to the noise introduced by the acquisition device, which affects the extraction of the fluorescence signal. In accordance with the image characteristics, the mathematical morphology method was used to filter out the noise. First, open operation was performed to smoothen the image contour, cut thin lines, remove edge burrs and outliers, and polish the image outer boundary. Then, close operation was performed to connect the shorter breakpoints in the image, fill the small gaps and smoothen the inner edges of the image.

Given that the C line and the T line are on the different sides of the strip, image processing can be divided into left and right parts, and the Otsu segmentation algorithm based on image entropy was adopted for them, respectively [20]. The algorithm is summarized as follows:

The image was assumed to have L gray levels. The probability of each gray value is *p*_i_. The threshold *t* divides different gray values into target class A (gray values in the range of 0–t) and background class B (gray values in the range of t + 1 − L − 1).

The information entropy of the target area is
(7)$$H_{A}=-\sum_{i=0}^{t}p_{i}\log_{2}p_{i}.$$
The information entropy of the background area is
(8)$$H_{B}=-\sum_{i=t+1}^{L-1}p_{i}\log_{2}p_{i}.$$
The entropy of the entire image is
(9)$$H_{0}=-\sum_{i=0}^{L-1}p_{i}\log_{2}p_{i}.$$
In this algorithm, the entropy function is used to replace the gray mean value in Otsu criterion, and the improved Otsu formula is as follows:(10)$$t^{*}=\mathrm{Arg}\underset{0<t<L-1}{\mathrm{Max}}\left[\frac{{\left(H_{A}-H_{0}\right)}^{2}{\left(H_{B}-H_{0}\right)}^{2}}{{\left({\left(H_{A}-H_{0}\right)}^{2}+{\left(H_{B}-H_{0}\right)}^{2}\right)}^{2}}\right],$$
where *t** is the best threshold; Figure 4 presents the program flow chart of image segmentation.

Figure 5 shows the image signal and segmentation results. This improved method has good self-adaptability and is unaffected by the linear change of gray value (image contrast change) and shift change (image brightness change) [20].

#### 2.2.2. Range Adjustment

For digital images, the gray level of the image is limited, and too high or too low gray value of the image leads to the FICA reader being unable to extract the correct information of the strip. Equation (2) shows that the photometric exposure of the image is proportional to the light intensity and exposure time. Therefore, the gray value of the strip image can be controlled within the detectable range by changing the light intensity and exposure time. The light intensity can be increased by controlling the current value to enhance the weak but useful signal [15]. However, the fluorescent substances of the strip will produce a photobleaching phenomenon as the light intensity exceeds a certain limit [21]. In this case, the current value is not linear with the excitation light intensity and the reader works in a non-linear region [15]. This study controlled the light intensity at a low fixed value by adjusting the current value to reduce photobleaching and finally implemented the automatic range adjustment by controlling the exposure time.

When a low-concentration strip is detected, the fluorescent signal excited by the strip is weak. After image processing, such as acquisition and filtering, useful signals may be submerged, and the detection program cannot easily extract useful information with accuracy. The system increases the photometric exposure by extending the exposure time of the CMOS image sensor to control the strip within the detectable range. Similarly, when a high-concentration strip is detected, the gray value of the image extracted by the detection program is limited and cannot correctly indicate the concentration information of the strip. At this point, the system performs the adjustment opposite to the detection of a low-concentration strip. Therefore, the system can automatically adjust the exposure time according to the concentration of strips under test to achieve the effect of automatic range adjustment. Figure 6 demonstrates the adjustment method. 

To maximize the detection range of the FICA reader based on image processing, a range control function was established to adjust the exposure time of the CMOS image sensor, so that the gray mean value of the target area reached the maximum (assumed to be *G*_M_) allowed by the detection system. The gray value *G* of the target area was calculated when the exposure time was *t*. Then the final exposure time was set as
(11)$$t_{M}=Kt,$$
where *K* is the ratio of the maximum gray value to the current target gray value (*G*_M_/*G*). Thus, the target gray value reached the maximum within the detectable range, and the range adjustment of the FICA reader was implemented.

## 3. Results

### 3.1. Range Adjustment Effect of FICA Strips

The effect of automatic range adjustment of the FICA reader based on image processing is shown in Figure 7 and Figure 8.

### 3.2. Range Comparison

C-reactive protein (CRP) solution for bacterial or viral infection identification was selected as the reagent to extract the characteristic values. The standard CRP solution was diluted to 0.98, 1.95, 3.91, 7.81, 15.6, 31.25, 62.5, 125, 256, 512 and 1024 μg/mL. 75 μL of the prepared solution was dropped into PBS buffer provided by the CRP kit (Triplex International Biosciences Co., LTD., Xiamen, China) and mixed evenly, then the mixed solution was dropped into the sample hole of the strip. CRP strips were labeled with fluorescent particles, with a detection sensitivity of 0.5 μg/mL and a reaction time of 3 min. After the reaction was completed, the ESEQuant Lateral Flow Reader (LFR) (Qiagen Co., LTD., Dusseldorf, Germany) and the FICA reader based on image processing were used to extract the characteristic value respectively. CRP solution of each solubility was measured 3 times, and the mean was taken as the final characteristic value. Table 1 presents the comparison results.

Table 1 shows that the maximum detection range has been expanded from 125 to 512 μg/mL by using automatic range adjustment.

### 3.3. Linearity

The detection results were fitted using the least squares method. Figure 9 presents the fitting results.

The linear relationship is
(12)y=0.023x+0.081,
where *y* is equal to the characteristic value *T*/*C*, and *x* is the concentration of the solution under test. The fitting degree (*R*^2^) of the fitted curve reaches 0.999, indicating that the linearity of the reader is satisfactory within the detectable range.

### 3.4. Accuracy

This study utilized ESEQuant LFR as a reference to verify the accuracy of the reader. Despite the high precision of ESEQuant LFR, its range is limited. Therefore, we linearly extended the detection results of ESEQuant LFR to the range of our reader. As shown in Figure 10, curve fitting was performed between the detection results of the reader and ESEQuant LFR. The fitting degree (*R*^2^) of the fitted curve reaches 0.999, which indicates the accuracy of the detection results of the FICA reader.

## 4. Discussion

Current research on FICA mainly focuses on the use of instruments in various fields, rather than the improvement of instrument performance. In order to reduce the cost and further popularize the instrument, it is necessary to extend the range of FICA reader using common image sensors.

This work described the quantitative detection and exposure time adjustment mechanisms of FICA strips. The change in excitation light intensity and exposure time had no effect on the detection result *T*/*C*. A FICA reader based on image processing was designed, and a method was proposed to increase the detection range by adjusting the exposure time. This method applies not only to FICA readers, but also for colloidal gold and other ICA devices.

This experiment used strips with different CRP solution concentrations as test samples. The results showed that the range of the reader before adjustment is 1.95–125 μg/mL, and the detection range after adjustment was expanded to 0.98–512 μg/mL. In addition, the upper detection limit of the reader was considerably higher than that of ESEQuant LFR. After the range adjustment, the fitting degree (*R*^2^) of the fitted curves in the linearity and accuracy experiments both reached 0.999. Therefore, the automatic range adjustment method not only effectively improves the detection sensitivity and upper detection limit of the detector, but also has good consistency with the detection results of the ESEQuant LFR.

The next step is to develop a detection system on the mobile platform. This procedure eliminates the dependence of the reader on PC and reduces the volume to gain further improvement in the portability of the reader.

## Figures and Tables

**Figure 1 sensors-20-00209-f001:**
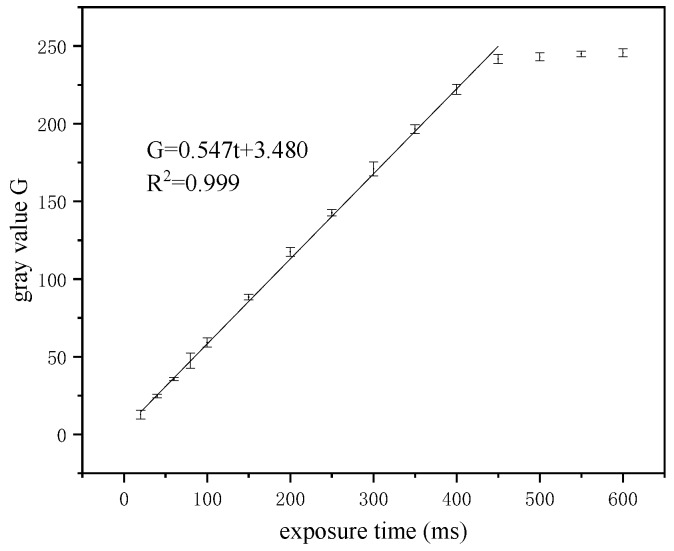
Relationship between the image gray value and exposure time.

**Figure 2 sensors-20-00209-f002:**
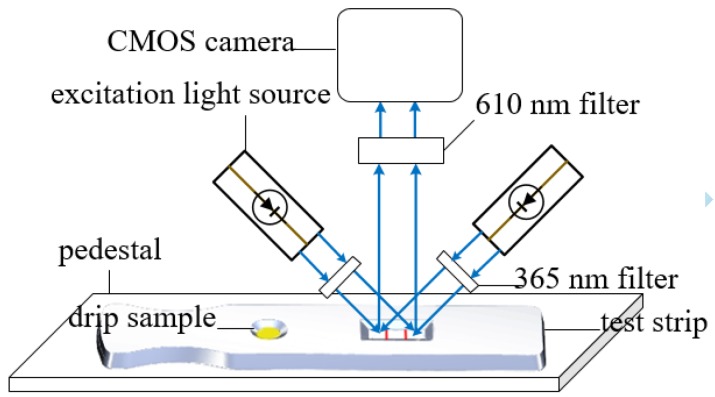
Image acquisition device of the fluorescence immunochromatographic assay (FICA) reader based on image processing.

**Figure 3 sensors-20-00209-f003:**
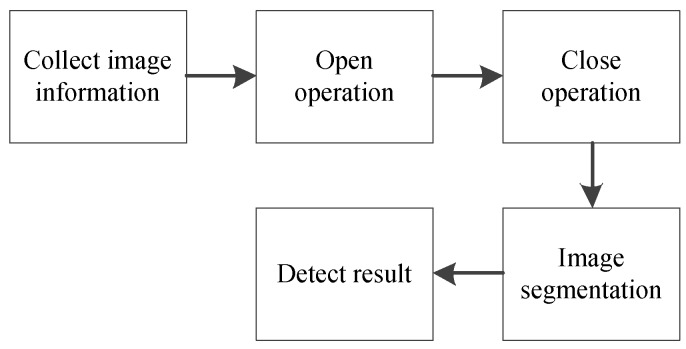
Image processing flowchart of FICA strips.

**Figure 4 sensors-20-00209-f004:**
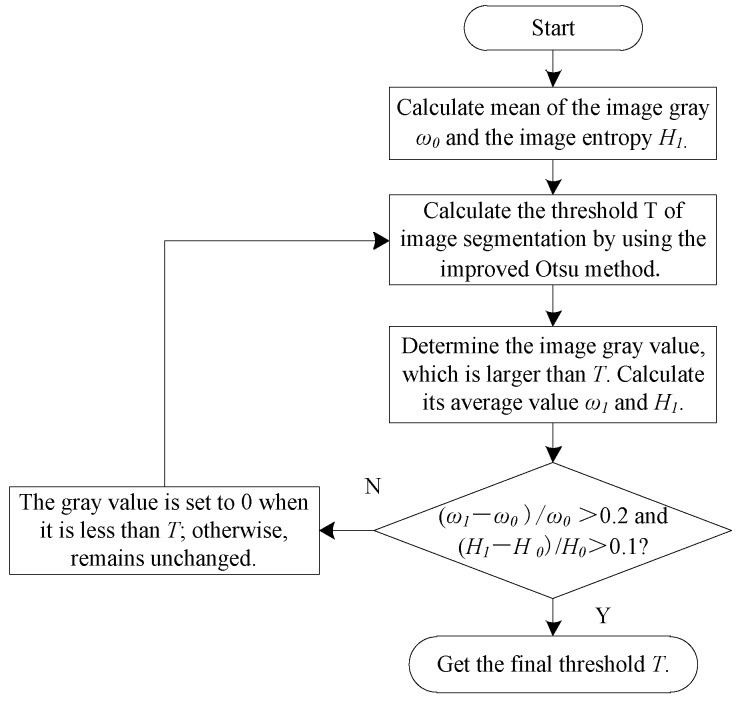
Flow chart of Otsu segmentation algorithm based on image entropy.

**Figure 5 sensors-20-00209-f005:**
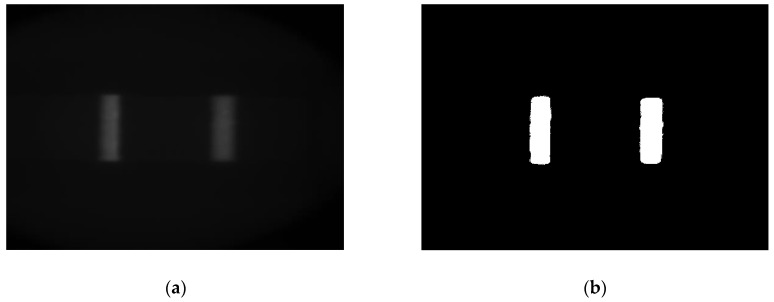
Results of image segmentation: (**a**) original image; (**b**) segmented image.

**Figure 6 sensors-20-00209-f006:**
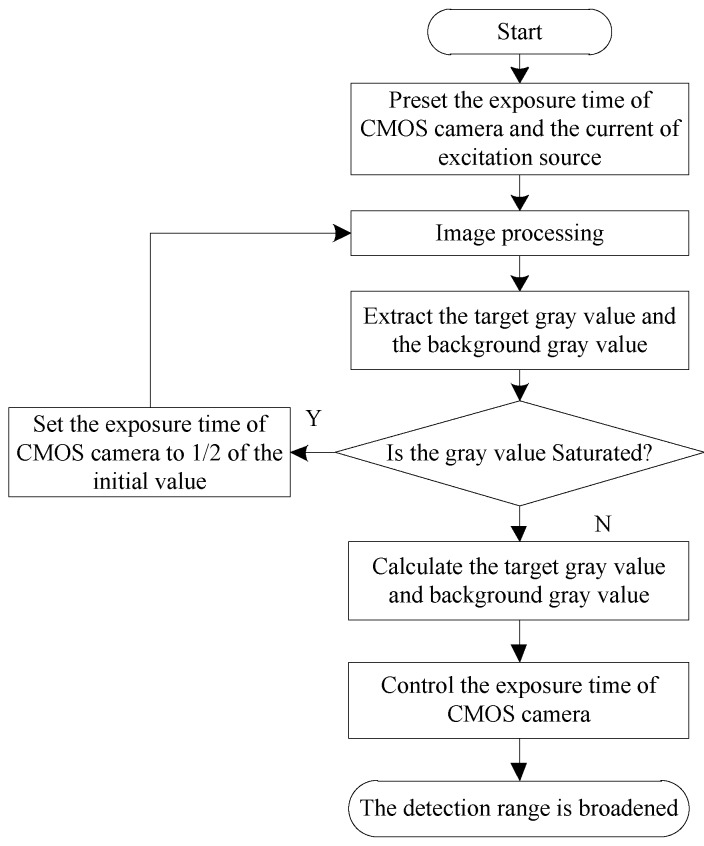
Flowchart of automatic range adjustment.

**Figure 7 sensors-20-00209-f007:**
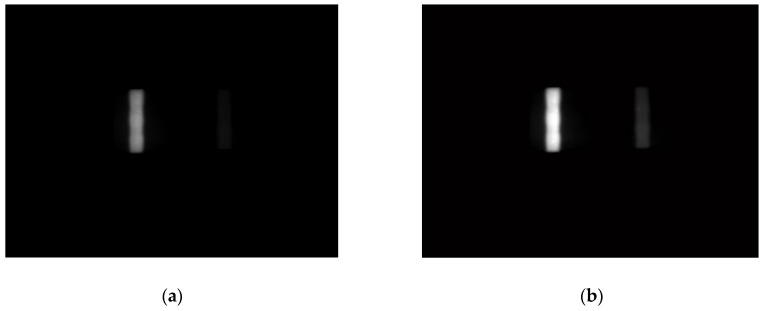
Detection result of a low-concentration strip: (**a**) original image; (**b**) after range adjustment.

**Figure 8 sensors-20-00209-f008:**
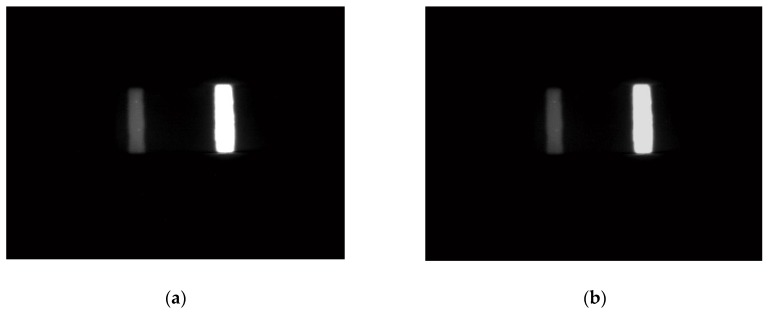
Detection result of a high-concentration strip: (**a**) original image; (**b**) after range adjustment.

**Figure 9 sensors-20-00209-f009:**
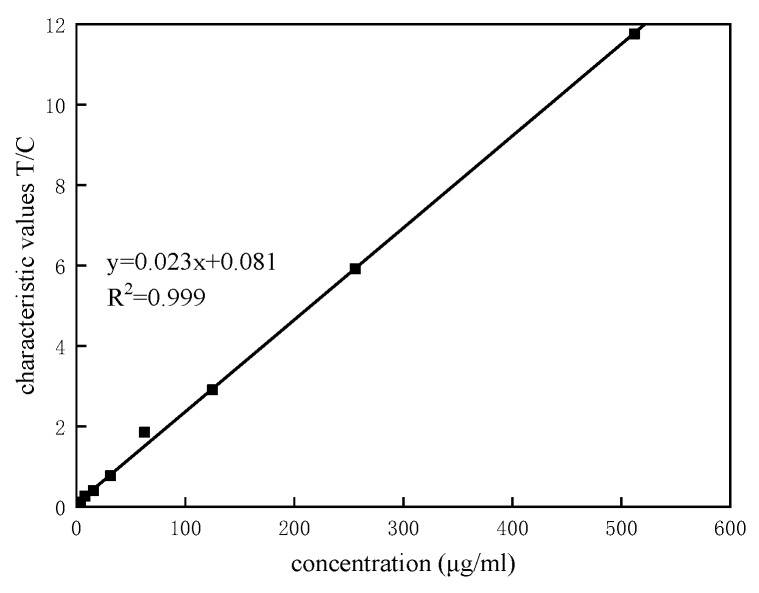
Linearity of the detection results after automatic range adjustment.

**Figure 10 sensors-20-00209-f010:**
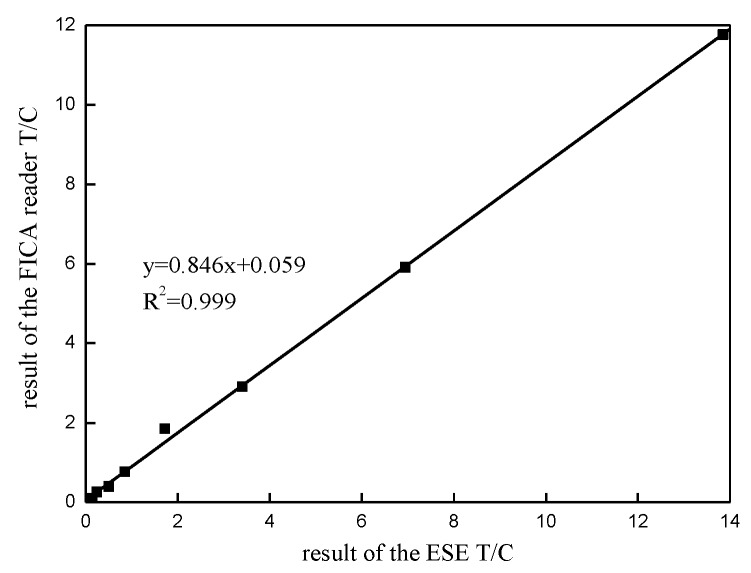
Correlation of detection results between ESEQuant LFR and the FICA reader after automatic range adjustment.

**Table 1 sensors-20-00209-t001:** Comparison of detection ranges of C-reactive protein (CRP) solution between the ESEQuant Lateral Flow Reader (LFR) and the FICA reader after automatic range adjustment.

CRP (μg/mL)	ESEQuant LFR	FICA Reader without Adjustment	FICA Reader with Adjustment
0.98	—	—	0.03
1.95	0.05	0.05	0.05
3.91	0.14	0.10	0.11
7.81	0.24	0.25	0.26
15.6	0.5	0.41	0.40
31.25	0.85	0.75	0.77
62.5	—	1.84	1.85
125	—	2.92	2.91
256	—	—	5.92
512	—	—	11.76
1024	—	—	—

Note: “—” Indicates that the result is invalid.

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
