# Peer review of "Automatic Range Adjustment of the Fluorescence Immunochromatographic Assay Based on Image Processing"

_sensors, 2019, doi:10.3390/s20010209_

Round 1

Reviewer 1 Report

The objectives of the study are not formulated.
Disadvantages of existing systems were indicated in the introduction. However, they were not affected in the study, or are not correct. For example, “FICA reader needs a mechanical scanning system” is indicated, but the article does not address this topic. Specifies "the reader only contains 1D information". This is not so. In all devices, two-dimensional information (from CMOS) is translated into a one-dimensional more convenient integrated graph. "the image processing of FICA cannot easily meet the requirements of wide detection range" is unconfirmed.

The article contains a lot of “trivial data” that are not always related to research. For example, "The collected image was transmitted to the upper computer by the USB interface" (p2 line 71). And the entire section of line 66-106 is given for the conclusion that the ratio of the intensity of the control and test zones is equal to the ratio of the fluorescent marker in the control and analytical zones.

No data on the test strips themselves, markers, or analysis is provided. There are no real samples.

The result of the study is modest.
A 2-fold increase in sensitivity for immunochromatography is not significant. Moreover, if you take the same picture 7a and contrast it in Photoshop, you can easily see the same result or even better.
The change in the upper range seems insignificant. Images 8a and 8b do not differ from each other.

Author Response

Reviewer#1, Concern # 1: The objectives of the study are not formulated.

Author response: Firstly, thanks for your attention. Secondly, we agree with your suggestion that the objectives of the study are not formulated.

Author action: We have revised the abstract to reflect our research objectives.

Reviewer#1, Concern # 2: Disadvantages of existing systems were indicated in the introduction. However, they were not affected in the study, or are not correct.

Author response: Firstly, thanks for your attention. Secondly, we agree with your suggestion that we have pointed out the disadvantages of the existing system, but we have not made analysis or are not correct.

Author action: We re-analyzed the deficiencies of the existing FICA readers and further highlighted our research objectives in the introduction section.

Reviewer#1, Concern # 3: The article contains a lot of “trivial data” that are not always related to research.

Author response: Firstly, thanks for your attention. Secondly, we agree with your suggestion that "trivial data" be included in this article.

Author action: We found that the first paragraph of section 2.1.1 made some irrelevant description, so we deleted part of it to make the article more concise. For the conclusion that more text is given to get the final detection result T/C equals CT / CC and has nothing to do with the exposure time of the camera, it is because this is the premise that we can adjust the range by adjusting the exposure time.

Reviewer#1, Concern # 4: No data on the test strips themselves, markers, or analysis is provided. There are no real samples.

Author response: Firstly, thanks for your attention. Secondly, we agree with your suggestion that no data on the test strips themselves, markers, or analysis is provided.

Author action: We didn’t introduce the test strip before, so we re-described the test strip in the range comparison section. At present, we mainly focus on the development of image fluorescence detection equipment. We use standard samples to test and have been able to reflect the performance of the instrument.

Reviewer#1, Concern # 5: The result of the study is modest. A 2-fold increase in sensitivity for immunochromatography is not significant. The change in the upper range seems insignificant.

Author response: Thanks for your attention.

Author action: We use a common 8-bit CMOS image sensor, so our expansion ability may be limited. However, after the range adjustment, its detection range is 0.98-512 μg/mL, which meets the detection range of common biochemical substances.

Reviewer#1, Concern # 6: If you take the same picture 7a and contrast it in Photoshop, you can easily see the same result or even better. Images 8a and 8b do not differ from each other.

Author response: Thanks for your attention.

Author action: We mentioned in the section of mechanism of exposure time adjustment and range adjustment that in order to maximize the detection range, we adjusted the exposure time to maximize the gray value of the target area. As can be seen from Fig. 7(b) and Fig. 8(b), the gray value of the adjusted image is very close. In the process of adjusting the exposure time, we can adjust it automatically through the program without manual adjustment in Photoshop. It can also be seen in Figure 1 that the maximum gray value is around 240, which is close to the maximum gray value of 255. Therefore, for figure 8(a), whose gray value exceeds 240, the adjustment process only slightly reduces the gray value, so it looks very close to Fig. 8(b). Moreover, our final results are not mainly identified by eyes, but obtained from the upper computer.

Reviewer 2 Report

X. Jiang et al. described a novel quantitative detection and exposure time adjustment mechanisms of fluorescence immunochromatographic assay (FICA) strips. The change in excitation light intensity and exposure time had no effect on the detection result T/C. A FICA reader based on image processing was designed, and a method was proposed to increase the detection range by adjusting the exposure time. This method is not only applied to FICA readers, but also for colloidal gold and other ICA devices. In view of development trend of point-of-care testing, FICA researches may attract many readers. However, this manuscript needs much careful major revision before it’s published.

The abstract needs to be rewritten. The list of Objective, Methods, Results, Conclusion is not a common abstract in Sensors. There are many grammar errors which should be revised carefully. For example, in the line 41-44 of the manuscript., ‘There are three kinds of biomarkers in the ICA applications, including nano-gold, quantum-dot, and fluorescent particles. Among them, fluorescence immunochromatographic assay (FICA) has’; FICA is not the biomarkers. Statement is ambiguous. In abstract and line 112, ‘goodness of fit R2’ is not an accurate expression.  In line 58, the authors claim ‘The present work studied in depth the shortcomings of FICA’. However, there is no enough evidence for that in this manuscript, please revise the description. 4. In table 1, CRP is presenting 0.98 μg/mL; but in conclusion, it shows ‘expanded to 1–512 μg/mL’. The description of language is not strict. Please supply the error bar of figure 1 Relationship between the image gray value and exposure time.

Author Response

Reviewer#2, Concern # 1: The abstract needs to be rewritten.

Author response: Firstly, thanks for your attention. Secondly, we agree with your suggestion that the abstract needs to be rewritten.

Author action: We have rewritten the abstract as required.

Reviewer#2, Concern # 2: There are many grammar errors which should be revised carefully.

Author response: Firstly, thanks for your attention. Secondly, we agree with your suggestion that FICA is not the biomarkers

Author action: We have modified the introduction of ICA and FICA in the introduction section and highlighted the revised parts of the whole article with red marks.

Reviewer#2, Concern # 3: Statement is ambiguous. In abstract and line 112, ‘goodness of fit R2’ is not an accurate expression.

Author response: Firstly, thanks for your attention. Secondly, we agree with your suggestion that our expression of ‘goodness of fit’ in abstract and line 112 is not accurate enough

Author action: We have made corresponding modifications in the abstract and the text, and marked them in red.

Reviewer#2, Concern # 4: In line 58, the authors claim ‘The present work studied in depth the shortcomings of FICA’. However, there is no enough evidence for that in this manuscript, please revise the description.

Author response: Firstly, thanks for your attention. Secondly, we agree with your suggestion that there is no enough evidence for that.

Author action: We re-analyzed the shortcomings of the existing FICA reader and revised the description in the introduction section.

Reviewer#2, Concern # 5: In table 1, CRP is presenting 0.98 μg/mL; but in conclusion, it shows ‘expanded to 1–512 μg/mL’. The description of language is not strict.

Author response: Thanks for your attention. Secondly, we agree with your suggestion that the description of language is not strict.

Author action: I'm sorry that we didn't make a strict statement before. We have modified it in the discussion section of the article.

Reviewer#2, Concern # 6: Please supply the error bar of figure 1 Relationship between the image gray value and exposure time.

Author response: Thanks for your attention.

Author action: We have provided the error bar diagram of figure 1 as required

Round 2

Reviewer 1 Report

All is ok.

Reviewer 2 Report

The authors responded well to the related questions and have revised the manuscript. In view of the importance of the relevant field, the manuscript is recommended to be published.